

# Worldwide inequality in access to full text scientific articles: the example of ophthalmology

Christophe Boudry[1,2], Patricio Alvarez-Muñoz[3],
Ricardo Arencibia-Jorge[4], Didier Ayena[5], Niels J. Brouwer[6],
Zia Chaudhuri[7], Brenda Chawner[8], Emilienne Epee[9], Khalil Erraïs[10],
Akbar Fotouhi[11], Almutez M. Gharaibeh[12], Dina H. Hassanein[13],
Martina C. Herwig-Carl[14], Katherine Howard[15],
Dieudonne Kaimbo Wa Kaimbo[16], Patricia-Ann Laughrea[17],
Fernando A. Lopez[18], Juan D. Machin-Mastromatteo[19],
Fernando K. Malerbi[20], Papa Amadou Ndiaye[21], Nina A. Noor[22],
Josmel Pacheco-Mendoza[23], Vasilios P. Papastefanou[24],
Mufarriq Shah[25], Carol L. Shields[26], Ya Xing Wang[27], Vasily Yartsev[28]
and Frederic Mouriaux[29,30]

[1] Normandie Univ, UNICAEN, Média Normandie, Caen, France
[2] URFIST, Ecole Nationale des Chartes, PSL Research University, Paris, France
[3] Universidad Estatal de Milagro, Milagro, Ecuador
[4] Empresa de Tecnologías de la Información (ETI), Grupo de las Industrias Biotecnológica y Farmacéutica (BioCubaFarma), Havana, Cuba
[5] Université de Lomé, Faculté des Sciences de la santé, Hôpital de Bè, Lomé, Togo
[6] Leiden University Medical Center, Department of Ophthalmology, Leiden, The Netherlands
[7] University of Delhi, Lady Hardinge Medical College, PGIMER, Dr RML Hospital, New Delhi, India
[8] Victoria University of Wellington, School of Information Management, Wellington, New Zealand
[9] Université de Yaoundé, Faculté de Médecine et des Sciences Biomédicales, Yaoundé, Cameroon
[10] Université de Tunis El-Manar, Faculté de Médecine de Tunis, Tunis, Tunisia
[11] Tehran University of Medical Sciences, School of Public Health, Department of Epidemiology and Biostatistics, Tehran, Iran
[12] University of Jordan, Faculty of Medicine, Amman, Jordan
[13] Cairo University, Ophthalmology Department, Cairo, Egypt
[14] University of Bonn, Department of Ophthalmology, Bonn, Germany
[15] Flinders University, Adelaide, Australia
[16] University of Kinshasa, Kinshasa, Democratic Republic of the Congo
[17] Laval University, Department of Ophthalmology and Head and Neck Surgery, Quebec, Canada
[18] Universidad Metropolitana para la Educación y el Trabajo, Centro de Innovación de los Trabajadores, Consejo Nacional de Investigaciones Científicas y Técnicas, Buenos Aires, Argentina
[19] Universidad Autónoma de Chihuahua, Chihuahua, Mexico
[20] Hospital Israelita Albert Einstein, Sao Paulo, Brazil
[21] Cheikh Anta Diop University, Abass NDAO Hospital, Dakar, Senegal
[22] JEC Eye Hospitals and Clinics, Jakarta, Indonesia
[23] Unidad de Investigación en Bibliometría, Universidad San Ignacio de Loyola, Lima, Peru
[24] Royal London Hospital, Barts Health NHS Trust. Moorfields Eye Hospital, London, UK
[25] Pakistan Institute of Community Ophthalmology, Hayatabad Medical Complex, Department of Optometry, Peshawar, Pakistan
[26] Wills Eye Hospital, Thomas Jefferson University, Philadelphia, PA, USA
[27] Beijing Institute of Ophthalmology, Beijing Tongren Hospital, Capital Medical University, Beijing Ophthalmology and Visual Sciences Key Laboratory, Beijing, China
[28] Scientific Research Institute of Eye Diseases, Moscow, Russia
[29] Univ Rennes, INSERM, INRA, CHU de Rennes, Institut NUMECAN (Nutrition Metabolisms and Cancer), Rennes, France
[30] Service d'ophtalmologie, CHU de Rennes, Rennes, France

Corresponding author
Christophe Boudry,
christophe.boudry@chartes.psl.eu

## ABSTRACT

**Background:** The problem of access to medical information, particularly in low-income countries, has been under discussion for many years. Although a number of developments have occurred in the last decade (e.g., the open access (OA) movement and the website Sci-Hub), everyone agrees that these difficulties still persist very widely, mainly due to the fact that paywalls still limit access to approximately 75% of scholarly documents. In this study, we compare the accessibility of recent full text articles in the field of ophthalmology in 27 established institutions located worldwide.

**Methods:** A total of 200 references from articles were retrieved using the PubMed database. Each article was individually checked for OA. Full texts of non-OA (i.e., "paywalled articles") were examined to determine whether they were available using institutional and Hinari access in each institution studied, using "alternative ways" (i.e., PubMed Central, ResearchGate, Google Scholar, and Online Reprint Request), and using the website Sci-Hub.

**Results:** The number of full texts of "paywalled articles" available using institutional and Hinari access showed strong heterogeneity, scattered between 0% full texts to 94.8% (mean = 46.8%; SD = 31.5; median = 51.3%). We found that complementary use of "alternative ways" and Sci-Hub leads to 95.5% of full text "paywalled articles," and also divides by 14 the average extra costs needed to obtain all full texts on publishers' websites using pay-per-view.

**Conclusions:** The scant number of available full text "paywalled articles" in most institutions studied encourages researchers in the field of ophthalmology to use Sci-Hub to search for scientific information. The scientific community and decision-makers must unite and strengthen their efforts to find solutions to improve access to scientific literature worldwide and avoid an implosion of the scientific publishing model. This study is not an endorsement for using Sci-Hub. The authors, their institutions, and publishers accept no responsibility on behalf of readers.

**Subjects** Ophthalmology, Ethical Issues, Legal Issues, Science Policy
**Keywords** Science publishing, Sci-Hub, Paywall, Bibliodiversity, Hinari, PubMed Central, ResearchGate, Google Scholar, Online Reprint Request, Pay-per-view, Ophthalmology, Access to literature, Articles, Open access

## INTRODUCTION

High-quality information is essential for effective health systems as well as scientific progress and development (*UNESCO, 1997*; *Koehlmoos & Smith, 2011*). Access to information in order to facilitate adequate health care is also considered to be a human right (*Goehl, 2007*; *United Nations, 2015*). On the contrary, the lack of access to knowledge is the main limitation to human development (*The Lancet, 2011*) and the principal barrier to knowledge-based health care in developing countries (*Godlee et al., 2005*).

Research studies are based on bibliographic research work to achieve "the state of the art." This stage not only makes it possible to carry out research studies based on existing and well established scientific foundations, but also prevents carrying out studies already conducted by other researchers in the world. Thus, access to scientific articles for researchers around the world is crucial for assessing high-quality research. In the current revenue-models of scientific journals, access to scientific papers is often restricted by paywalls. This implies that full text articles are only available upon subscription or pay-per-view on publisher websites. One is forced to note that paywalls still limit access to approximately 75% of scholarly documents in all disciplines (*Bosman & Kramer, 2018*; *Piwowar et al., 2018*), including biology and medicine (*Boudry & Durand-Barthez, 2017*; *Bosman & Kramer, 2018*). Moreover, "paywalled articles" are known to disadvantage the lowest-income countries in terms of access to articles (*Aronson, 2004*; *Himmelstein et al., 2018*).

Several initiatives have been implemented worldwide to facilitate access to scientific literature. For researchers in developing countries, a specific program called Hinari (Health InterNetwork Access to Research Initiative) was launched in 2002 for medicine. Hinari was developed by the World Health Organization (http://www.who.int/hinari/en/) and is now part of one of the five Research4Life programs created to reduce the knowledge gap in developing countries. It provides free or very low-cost online access to resources in the biomedical literature for not-for-profit institutions in developing countries, based on socio-economic factors (*World Health Organization (WHO), 2018*). In 2011, in response to a lack of access to scientific articles in Kazakhstan (*Bohannon, 2016*), Alexandra Elbakyan launched website Sci-Hub, allowing direct downloading of articles, bypassing publisher paywalls. A recent study showed that Sci-Hub contains 85.1% of all articles published in subscription-based journals (*Himmelstein et al., 2018*), and is extensively used worldwide in low-, middle- but also in high-income countries to circumvent paywalls (*Bohannon, 2016*). Such means of obtaining full text articles raises many legal (*Kemsley, 2017*; *Greco, 2017*) and ethical questions (*Bendezú-Quispe et al., 2016*; *Saleem, Hasaali & Ul Haq, 2017*; *Hoy, 2017*).

Data on the availability of scientific literature are often obtained from single-nation studies (*Nisonger, 2011*; *Malapela & De Jager, 2017*), and to the best of our knowledge, worldwide comparisons have not been carried out. Consequently, it is still not well known how paywalls may affect researchers worldwide. The primary objective of this study was to evaluate the accessibility of recent full text articles in the field of ophthalmology in established institutions distributed worldwide using "institutional and Hinari access," then "alternative ways" (PubMed Central, ResearchGate (RG), Google Scholar and Online Reprint Request (ORR)), and finally Sci-Hub. The secondary objective was to calculate extra costs institutions or researchers must bear to recover all full texts, buying them on publishers' websites.

## MATERIALS AND METHODS

The search for articles to be included in this study was carried out on March 29th, 2018 using the PubMed database (http://www.ncbi.nlm.nih.gov/pubmed), developed by the

National Center for Biotechnology Information at the National Library of Medicine. The search strategy was: eye diseases [MH] AND 2017:2018 [DP] AND English [LA] AND Journal Article [PT] where MH stands for "Medical Subject Headings," DP "Date of Publication," and PT "Publication Type." "Journal Article" includes the following publication types: journal articles, introductory journal articles, and reviews as previously described in the field of ophthalmology (*Boudry et al., 2016*). Data were downloaded from PubMed in Comma-Separate values and were imported to Microsoft Excel 2013 (Microsoft, Redmond, WA, USA) for data processing. This publication search resulted in a total of 11,103 articles recently published in the field of ophthalmology from which 200 (*Tenopir et al., 2010*) were randomly selected. The margin of error has been computed at 95% confidence level.

## Access to full text articles

Each of these 200 articles was individually checked for open access (OA) on the publisher's websites on April 3rd and 4th, 2018. Articles were labelled as "publisher OA" if the full text was accessible freely on the publisher's website without requiring subscription. Articles were labelled as "paywalled" if the full text was only available with a subscription.

The recruitment of the participants in the different institutions included in this study was done by searching recent articles (published in 2017 or 2018) in the field of ophthalmology. The researchers were contacted by email using corresponding authors' email addresses, or by contacting researchers already known by CB and FM directly. Only researchers working in capital cities or belonging to the most populated cities in their countries were chosen. Emails were sent to correspondents/colleagues to ask them to participate in this study, with the goal of reaching at least 25 countries located worldwide and spread over all continents.

Countries of researchers who agreed to collaborate were classified by continents according to the United Nations classification (*United Nations, 2018*) and classified by income level according to the World Bank classification (*World Bank, 2018*). Each participant had to search for "paywalled articles" in order to recover their full texts, using their institutional access and Hinari resources if available in their institutions, following a normalized protocol (see Supplemental Informations 1 and 2). The Hinari offer is available to two groups of countries: group A (free access for 69 countries) and group B (1,500 US dollars per calendar year for 50 countries) (*World Health Organization (WHO), 2018*).

Additionally, individual searches for all full text "paywalled articles" were done using "alternative ways" for evaluating the ability to find full texts of unavailable "paywalled articles" through institutional and Hinari access. These searches were done using the digital objects identifiers or titles of each article in the following order:

– Via PubMed Central open repository (https://www.ncbi.nlm.nih.gov/pmc/), the main free full-text open archive of biomedical and life sciences journal literature. Full text

articles on PubMed Central are legally deposited either by publishers or by authors themselves;

– Via the academic social network RG (http://www.researchgate.net/). Full text articles on RG are deposited by authors, sometimes without respecting publishing agreements and copyright laws (*Jamali, 2017*);

– Via Google Scholar (https://scholar.google.com/), looking at the first 10 results listed (*Nicholas et al., 2017*). Full texts retrieved via Google Scholar are those found by the search engine on websites throughout the internet, without any guarantee of legality regarding copyright laws;

– Via ORR. ORR use the corresponding author email to obtain non-open-access literature using reprints furnished by publishers to the corresponding author after publication of an article. ORR is thus fully legal.

Full texts retrieved via PubMed Central, RG, Google Scholar or ORR were labelled as "alternative ways."

– Via the Sci-Hub website. This means of obtaining full text is illegal in many countries in the world as full texts on Sci-Hub are pirated from legal websites such as libraries, without regard to copyright laws.

As the accessibility of full text "paywalled articles" using Sci-hub was independent of the geographical location where the search was done, searches for full text "paywalled articles" using Sci-hub were performed from France, from a non-university Internet access. No university or institution of the authors of this study was therefore involved in downloading articles via Sci-Hub.

This study has been approved by the Ethics Committee of the University Hospital of Rennes.

## Financial implications of unavailable full text articles

When researchers do not have access to full text "paywalled articles" through the different means of access at their disposal, they can buy them individually on publisher's websites (pay-per-view). However, this extra cost presents the disadvantage of being borne directly by the institution or laboratory, and sometimes by the researchers themselves. Extra costs of unavailable full text "paywalled articles" were calculated both in US dollars and as the percentage of gross domestic product (GDP) per capita at purchasing power parity (PPP). For each "paywalled article," the cost of the pay-per-view of each article was recovered from the publisher's website in US dollars. When the price was in euros, it was converted to US dollars using the exchange rate in use on April 5th, 2018. The cost of unavailable full text articles in each institution was calculated in US dollars, and was also expressed as the percentage of the GDP per capita at PPP (current international $), using the World Development Indicators from the online databases of the World Bank (World Bank, 2018). The GDP is the market value of all officially recognized final goods and services produced within a country in a given period. Using the percentage of

**Table 1 Institutions included in the study presented by continents (listed alphabetically).**

| Institution | Country | World bank classification by income level |
|---|---|---|
| University of Yaounde | Cameroon | Lower-middle |
| University of Kinshasa | Democratic Republic of the Congo | Lower-middle |
| Cairo University | Egypt | Lower-middle |
| University of Dakar | Senegal | Low |
| Lomé University | Togo | Low |
| University El-Manar, Tunis | Tunisia | Lower-middle |
| Laval University | Canada | High |
| Thomas Jefferson University, Philadelphia | United States of America | High |
| Universidad Metropolitana para la Educación y el Trabajo, Buenos Aires | Argentina | Upper-middle |
| Hospital Israelita Albert Einstein, Sao Paulo | Brazil | Upper-middle |
| Empresa de Tecnologías de la Información, Havana | Cuba | Upper-middle |
| State University of Milagro | Ecuador | Upper-middle |
| Autonomous University of Chihuahua | Mexico | Upper-middle |
| National University of San Marcos, Lima | Peru | Upper-middle |
| Capital Medical University, Beijing | China | Upper-middle |
| University of Delhi | India | Lower-middle |
| Jakarta Eye Center | Indonesia | Lower-middle |
| Tehran University of Medical Sciences | Islamic Republic of Iran | Upper-middle |
| University of Jordan | Jordan | Lower-middle |
| Pakistan Institute of Community Ophthalmology, Peshawar | Pakistan | Lower-middle |
| Pierre and Marie Curie University, Paris | France | High |
| University of Bonn | Germany | High |
| Leiden University Medical Center | The Netherlands | High |
| Eye Diseases Research Institute, Moscow | Russian Federation | Upper-middle |
| Moorfields Eye Hospital, London | United Kingdom | High |
| Flinders University, Adelaide | Australia | High |
| Victoria University of Wellington | New Zealand | High |

GDP per capita at PPP allows us to assess the real financial burden of providing full text articles in relation to the standard of living of each country.

# RESULTS

We received 26 positive responses to the 166 emails sent seeking colleagues for participation in this study. Table 1 describes institutions and characteristics of the 27 countries included in the present study. Four were Hinari group A (Cameroon, Democratic Republic of Congo, Senegal, and Togo), whereas four were Hinari group B (Egypt, Tunisia, Jordan, and Pakistan) and 19 others were not Hinari eligible. Among institutions located in Hinari group B eligible countries, only the University El-Manar (Tunisia) applied to obtain Hinari resources.

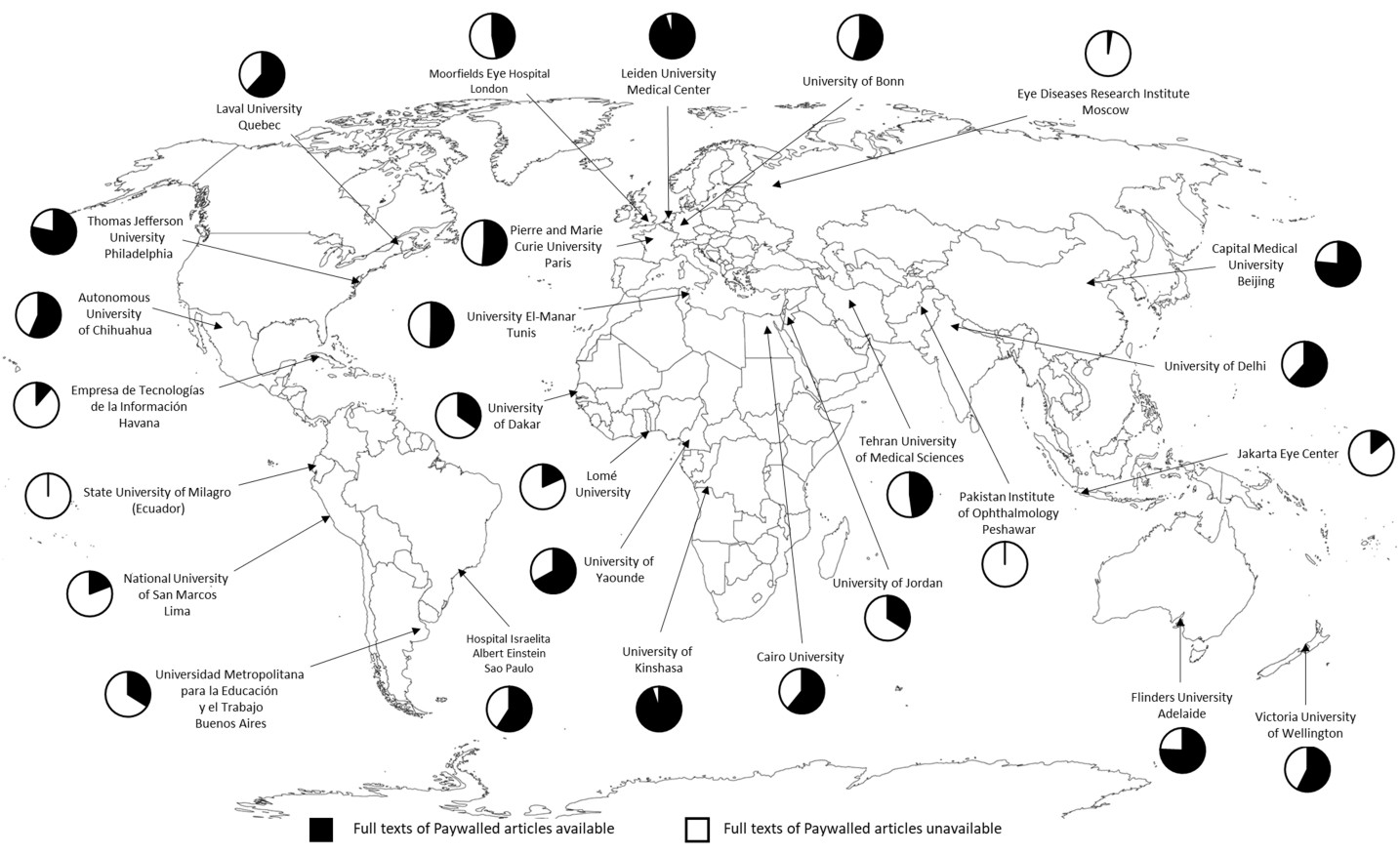

**Figure 1** Number of full text "paywalled articles" available ($n = 115$) for each of the 27 institutions included in the study.

## Access to full text articles

Among the 200 articles studied, 85 full texts (42.5%) were freely available on publishers' websites ("publisher OA" articles), whereas 115 (57.5%) full texts were available only with subscriptions ("paywalled articles"). See Supplemental Information 3 for the list of the 115 "paywalled articles."

Figure 1 presents the number of full texts of the 115 "paywalled articles" available in each institution using institutional and Hinari access. It must be emphasized that the number of available full texts using institutional and Hinari access showed substantial heterogeneity, scattered between 0 (0%) full texts and 109 (94.8%). An average of 53.8 (46.8%) full text "paywalled articles" (SD = 31.5) were available (Table 2). Half of the institutions had access to less than 59 (51.3%) full text "paywalled articles" (IQR = 49).

Regarding "alternative ways," 82 e-mails of corresponding authors were found via PubMed or via the Publisher website, and ORRs were sent to them. A total of 31 responses were received (43.9% success rate), allowing us to obtain 26.96% of "paywalled articles" using ORRs. No full texts of paywalled articles were found on the open archive PMC (paywalled articles included in this study were very recent and still under publisher

**Table 2 Number of full text "paywalled articles" available using institutional and Hinari access, "alternative ways" (PubMeD Central, RG, Google Scholar, and ORR), and Sci-Hub.**

| Institution | Institutional/Hinari* n (%) | Institutional/Hinari* + "alternative ways" n (%) | Institutional/Hinari* + "alternative ways" + Sci-Hub n (%) |
|---|---|---|---|
| University of Yaounde (Cameroon) | 77 (67) | 95 (82.6) | 113 (98.3) |
| University of Kinshasa (Democratic Republic of the Congo) | 109 (94.8) | 111 (96.5) | 113 (98.3) |
| Cairo University (Egypt) | 70 (60.9) | 87 (75.7) | 110 (95.7) |
| University of Dakar (Senegal) | 40 (34.8) | 67 (58.3) | 111 (96.5) |
| Lomé University (Togo) | 21 (18.3) | 51 (44.3) | 108 (93.9) |
| University El-Manar, Tunis (Tunisia) | 58 (50.4) | 77 (67) | 112 (97.4) |
| Laval University, Quebec (Canada) | 71 (61.7) | 80 (69.6) | 109 (94.8) |
| Thomas Jefferson University, Philadelphia (United States of America) | 90 (78.3) | 99 (86.1) | 113 (98.3) |
| Universidad Metropolitana para la Educación y el Trabajo. Buenos Aires (Argentina) | 39 (33.9) | 60 (52.2) | 108 (93.9) |
| Hospital Israelita Albert Einstein, Sao Paulo (Brazil) | 68 (59.1) | 84 (73) | 109 (94.8) |
| Empresa de Tecnologías de la Información, Havana (Cuba) | 13 (11.3) | 41 (35.7) | 108 (93.9) |
| State University of Milagro (Ecuador) | 0 (0) | 37 (32.2) | 108 (93.9) |
| Autonomous University of Chihuahua (Mexico) | 65 (56.5) | 83 (72.2) | 110 (95.7) |
| National University of San Marcos, Lima (Peru) | 22 (19.1) | 50 (43.5) | 108 (93.9) |
| Capital Medical University, Beijing (China) | 88 (76.5) | 102 (88.7) | 111 (96.5) |
| University of Delhi (India) | 71 (61.7) | 80 (69.6) | 108 (93.9) |
| Jakarta Eye Center (Indonesia) | 16 (13.9) | 47 (40.9) | 108 (93.9) |
| Tehran University of Medical Sciences (Islamic Republic of Iran) | 55 (47.8) | 72 (62.6) | 108 (93.9) |
| University of Jordan (Jordan) | 39 (33.9) | 64 (55.7) | 109 (94.8) |
| Pakistan Institute of Community Ophthalmology, Peshawar (Pakistan) | 0 (0) | 37 (32.2) | 108 (93.9) |
| Pierre and Marie Curie University, Paris (France) | 59 (51.3) | 77 (67) | 111 (96.5) |
| University of Bonn (Germany) | 63 (54.8) | 75 (65.2) | 111 (96.5) |
| Leiden University Medical Center (The Netherlands) | 109 (94.8) | 111 (96.5) | 112 (97.4) |
| Eye Diseases Research Institute, Moscow (Russian Federation) | 3 (2.6) | 38 (33) | 108 (93.9) |
| Moorfields Eye Hospital, London (United Kingdom) | 54 (47) | 78 (67.8) | 111 (96.5) |
| Flinders University, Adelaide (Australia) | 87 (75.7) | 95 (82.6) | 111 (96.5) |
| Victoria University of Wellington (New Zealand) | 66 (57.4) | 80 (69.6) | 109 (94.8) |
| Mean | 53.8 (48.6) | 73.3 (63.7) | 109.8 (95.5) |
| Standard deviation | 31.5 | 22.2 | 1.8 |
| Min | 0 (0) | 37 (32.2) | 108 (93.9) |
| Max | 109 (94.8) | 111 (96.5) | 113 (98.3) |
| Median | 59 (51.3) | 77 (67) | 109 (94.8) |
| Inter quartile range (IQR) | 49 | 36 | 3 |

**Note:**
* If available. Institutions are sorted by continent, countries are listed alphabetically.

embargoes, prohibiting self-archiving in OA repositories), and only 13 (11.30%) were found on social academic networks and internet via Google Scholar.

To complete the search for unavailable full texts using institutional and Hinari access, researchers could use "alternative ways" (PubMeD Central, RG, Google Scholar and ORR).

As presented in Table 2, by using complementary "alternative ways," an average of 73.3 of the 115 (63.7%) full text "paywalled articles" were available (SD = 22.2). The range of the number of available full texts varied among the institutions studied from 37 (32.2% for the Pakistan Institute of Community Ophthalmology and the State University of Milagro) to 111 (96.5% for Leiden University Medical Center) of the 115 "paywalled articles." The margin of error, computed at 95% confidence level, was equal to 9.14%.

Used alone, Sci-hub allows the recovery of 108 full texts (93.9%) of 115. Despite its illegal nature, researchers may be tempted to use this website, to try to find the unavailable full texts using institutional/Hinari access and "alternative ways." Interestingly, using Sci-Hub as a complement to institutional/Hinari access and "alternative ways" allowed the recovery of an average of 109.8 (95.5%) full texts (SD = 1.8). Thus, the range of the number of available full texts varied very slightly from 108 (93.9%) to 113 (98.3%) of the 115 "paywalled articles." Moreover, for Pakistan Institute of Community Ophthalmology and State University of Milagro, complementary use of Sci-Hub allowed them to recover 108 full texts of 115 instead of 37. As shown in Fig. 2, when considering access by continent (Fig. 2A) or by income level (Fig. 2B), using complementary "alternative ways" tightens the gaps between continents and between countries of low and high income level. Using Sci-Hub as a complement to institutional/Hinari access and "alternative ways" totally eliminates the inequalities of access to "paywalled articles" by continent or income level. Note that the two institutions (University of Dakar and Lomé University) located in low income countries have particularly bad access despite using Hinari access. These findings suggest that Hinari program may not succeed in providing widespread access to less privileged countries.

## Financial implications of unavailable full text articles

Extra costs of unavailable full text "paywalled articles were assessed after using institutional/Hinari access, after using institutional/Hinari access and "alternative ways," and after using institutional/Hinari access, "alternative uses," and Sci-Hub (Table 3).

Using institutional/Hinari access, the average extra cost (in US dollars) of unavailable full text "paywalled articles" would be $2,790.8. However, this extra cost varies greatly from one institution to another: from $258 (Leiden University Medical Center) to $5,240.9 (State University of Milagro and Pakistan Institute of Community Ophthalmology). Using alternative ways in addition to institutional/Hinari access, extra costs would be reduced by an average of $1,918.6. Interestingly, complementary use of Sci-Hub reduces the average extra cost of unavailable full texts even more, dividing this average extra cost of unavailable full texts by 14 ($198.4 instead of $2,790.8 using institutional/Hinari access). Likewise, the average extra cost expressed as the percentage of GDP per capita at PPP is reduced more than 10-fold (2.4% instead of 25.4% using institutional/Hinari access

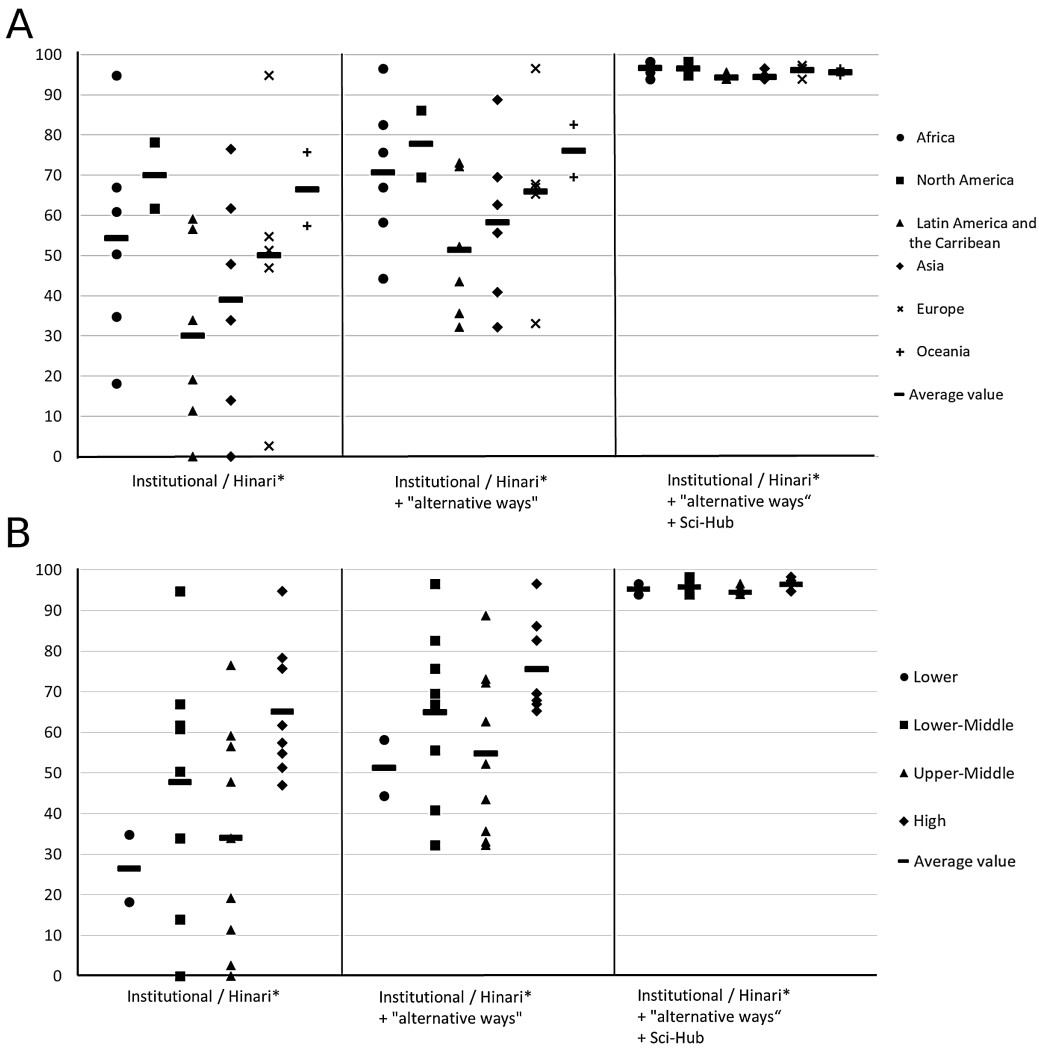

**Figure 2** Percentage of full text "paywalled articles" available by continent (A) and by income level (B) using institutional and Hinari access, "alternative ways" (PubMeD Central, RG, Google Scholar, and ORR), and Sci-Hub. *If available. 

and "alternative ways")  for the 27 institutions studied. This extra cost is thus substantial in some institutions, for example, in Togo, where it is at 18.8%.

## DISCUSSION

In order to describe the difficulty in accessing scientific literature, a number of studies only focusing on one country have been conducted, mostly in the USA, to assess the availability of journals or books in university libraries (*Nisonger, 2011*). Other studies have been conducted to analyze global access and use of digital resources in research, mainly in African universities (*Harle, 2009*; *Malapela & De Jager, 2017*; *Bruijns et al., 2017*). To the best of our knowledge, only one study in 2011 previously evaluated the accessing of recent full text "paywalled articles," and included seven institutions located in Africa, North and South America, Asia, and Europe (*Voronin, Myrzahmetov & Bernstein, 2011*). Furthermore, the problem of accessing medical information in low-income countries is

**Table 3 Extra cost of full text "paywalled articles" unavailable using institutional/Hinari access; using institutional/Hinari accesses and "alternative ways"; and using institutional/Hinari access and "alternative ways" and Sci-Hub.**

| Institution | Institutional/Hinari* | Institutional/Hinari* + "alternative ways" | Institutional/Hinari* + "alternative ways" + Sci-Hub |
|---|---|---|---|
| University of Yaounde (Cameroon) | 1,636.3 (45.3) | 799.0 (22.1) | 30 (0.8) |
| University of Kinshasa (Democratic Republic of the Congo) | 322.2 (5.6) | 165.9 (2.9) | 58 (1.0) |
| Cairo University (Egypt) | 1,940.2 (17.4) | 1,176.0 (10.6) | 162 (1.5) |
| University of Dakar (Senegal) | 3,475.1 (135.4) | 2,181.0 (85.0) | 190.8 (7.4) |
| Lomé University (Togo) | 4,378.1 (293.6) | 3,016.1 (202.3) | 279.6 (18.8) |
| University El-Manar, Tunis (Tunisia) | 2,638.3 (22.8) | 1,788.1 (15.4) | 160.8 (1.4) |
| Laval University, Quebec (Canada) | 2,296.4 (5.1) | 1,831.8 (4.1) | 235.6 (0.5) |
| Thomas Jefferson University, Philadelphia (United States of America) | 962 (1.7) | 579.4 (1.0) | 58 (0.1) |
| Universidad Metropolitana para la Educación y el Trabajo. Buenos Aires (Argentina) | 3,568.2 (17.9) | 2,644.8 (13.3) | 279.6 (1.4) |
| Hospital Israelita Albert Einstein, Sao Paulo (Brazil) | 2,035.2 (13.5) | 1,313.8 (8.7) | 220.8 (1.5) |
| Empresa de Tecnologías de la Información, Havana (Cuba) | 4,573.4 (84.0) | 3,391.6 (62.3) | 279.6 (5.1) |
| State University of Milagro (Ecuador) | 5,240.9 (47.0) | 3,597 (32.3) | 279.6 (2.5) |
| Autonomous University of Chihuahua (Mexico) | 2,116 (12.2) | 1,333.6 (7.7) | 162 (0.9) |
| National University of San Marcos, Lima (Peru) | 4,229.8 (32.5) | 3,034.6 (23.3) | 279.6 (2.1) |
| Capital Medical University, Beijing (China) | 1,150.7 (7.4) | 512.9 (3.3) | 118 (0.8) |
| University of Delhi (India) | 2,229.1 (33.9) | 1,778.5 (27.1) | 279.6 (4.3) |
| Jakarta Eye Center (Indonesia) | 4,657.6 (40.1) | 3,201.6 (27.6) | 279.6 (2.4) |
| Tehran University of Medical Sciences (Islamic Republic of Iran) | 2,706.9 (13.6) | 2,001.3 (10.0) | 279.6 (1.4) |
| University of Jordan (Jordan) | 3,446 (38.1) | 2,526.6 (27.9) | 279.6 (3.1) |
| Pakistan Institute of Community Ophthalmology, Peshawar (Pakistan) | 5,240.9 (100.1) | 3,597.0 (68.7) | 279.6 (5.3) |
| Pierre and Marie Curie University, Paris (France) | 2,608.4 (6.3) | 1,775.9 (4.3) | 176.8 (0.4) |
| University of Bonn (Germany) | 2,325.7 (4.8) | 1,749.7 (3.6) | 118.8 (0.2) |
| Leiden University Medical Center (The Netherlands) | 258 (0.5) | 124 (0.2) | 88 (0.2) |
| Eye Diseases Research Institute, Moscow (Russian Federation) | 5,134 (20.7) | 3,561.1 (14.4) | 279.6 (1.1) |
| Moorfields Eye Hospital, London (United Kingdom) | 2,579.8 (4.5) | 1,495.2 (2.6) | 118 (0.2) |
| Flinders University, Adelaide (Australia) | 1,270.4 (2.8) | 918.9 (2.0) | 146.8 (0.3) |
| Victoria University of Wellington (New Zealand) | 2,330.6 (6.0) | 1,707.9 (4.4) | 235.6 (0.6) |
| Mean | 2,790.8 (37.5) | 1,918.6 (25.4) | 198.4 (2.4) |
| SD | 1,458.2 (60.6) | 1,064.5 (41.5) | 83.7 (3.7) |
| Min | 258 (0.5) | 124 (0,2) | 30 (0.1) |
| Max | 5,240.9 (293.6) | 3,597 (202.3) | 279.6 (18.8) |

**Note:**

* If available. Extra cost indicates the sum of the price of unavailable full texts bought individually on publishers' websites in US dollars, or between brackets expressed as percentage gross domestic product (GDP) per capita at purchasing power parity (PPP) (current international $).

still being openly discussed (*Aronson, 2004*; *Godlee et al., 2005*; *Goehl, 2007*; *Himmelstein et al., 2018*). We believe that our study provides the most complete comparative study examining worldwide access to recent full text articles. In addition, it is the first to study

the financial implications of limited access to scientific literature and the first to assess Sci-Hub's performance as an alternative to legal institutional access. The field of ophthalmology was chosen because, for several years, our research team has been focusing its research on bibliometrics in the field of ophthalmology.

Our study nevertheless has a number of limitations. The number of countries studied is relatively low, particularly Hinari eligible countries, which can be explained by the difficulty in finding researchers in the field of ophthalmology in most of the Hinari group A and B eligible countries. Indeed, when we looked for collaborators to participate in this study, we found very few researchers in the field of ophthalmology located in these countries (personal data). Moreover, institutions studied in each country are inevitably not representative of the overall situation in the country, and requesting participation from all the institutions in a country is impossible. Nevertheless, to minimize the differences between institutions, only those located in capital cities or belonging to the most populated cities in their countries were chosen. Finally, it was not possible to obtain an average value of access to full text "paywalled articles" by country and to calculate correlations between the number of accessible full texts and socio-economic parameters (e.g., GDP per capita). For the same reasons, it was impossible to assess whether the use of Hinari resources significantly changed the number of available full text "paywalled articles." Furthermore, we wanted to compare the results obtained with the three ways of accessing full text "paywalled articles." Our data included the average number of full texts available and the average extra costs, in US dollars and expressed as the percentage of GDP per capita at PPP. We found that using institutional/Hinari access; using institutional/Hinari access and "alternative ways;" and using institutional/Hinari access, "alternative ways," and Sci-Hub were significantly different at $p < 0.05$ ($p$-value $< 0.001$). However, as the ways of accessing full text "paywalled articles" were not independent (because of the cumulative effect that binds them), these statistical results were not taken into account. We deliberately limited the search to articles published in 2017–2018 to include only very recent articles. This allowed us to access the latest findings which are thus presumed to be more informative for clinicians and researchers than older ones. Nevertheless, due to embargo periods, we must note that including articles published over a wider time period could have led to more frequent inclusion of articles freely available on publishers' websites ("publisher OA" articles) and articles using "alternative ways."

Our study nonetheless provides a global vision of the difficulty involved in accessing scientific literature around the world, highlighting the shortcomings of institutional access, which can have significant financial consequences for researchers seeking to overcome them. Our study also shows that the use of the website Sci-Hub helps to overcome these shortcomings and financial consequences.

The OA movement has been progressively making more articles openly available. However, articles behind paywalls ("paywalled articles") are still more numerous than "publisher OA" articles in our study (57.5% and 42.5%, respectively). These results are in agreement with the most recent studies, which estimate that, in biology and medicine, 39.1% to 50% of articles published are "publisher OA" articles (*Boudry & Durand-Barthez, 2017*; *Bosman & Kramer, 2018*). We found that researchers in ophthalmology working

in the 27 institutions included in this study can access only an average of 46.8% (with very large disparities, varying from 0% to 94.8%) of full text "paywalled articles" using their institutional and Hinari access. Furthermore, half of the 27 institutions studied can offer access to just over 50% of full text "paywalled articles," a value very close (56%) to that determined in 2011 for seven institutions located in Africa, North and South America, Asia, and Europe (*Voronin, Myrzahmetov & Bernstein, 2011*) showing that the situation has not improved in recent years. As previously pointed out by *Machin-Mastromatteo, Uribe-Tirado & Romero-Ortiz (2016)*, this shows that, "many universities are unable to acquire subscriptions for years, because they are seriously hindered by budget limitations and the lack of interest and policies from the state for supporting research and access to scientific resources." Researchers in ophthalmology working in the less privileged institutions are thus in very poor situations. Indeed, although they use "alternative ways" (i.e., PubMeD Central, RG, Google Scholar and ORR) to recover full texts, they are forced to use Sci-Hub if they want to access a sufficient number of full text articles necessary to conduct their research satisfactorily. Researchers cannot afford these articles because their yearly budget for accessing articles without using Sci-Hub is too high, with sums of money up to $3,597 (institutions in Pakistan or Ecuador), or more than 200% of the GDP per capita at PPP (202.3% in Ecuador).

Results of the present study showed that "alternative ways" might help researchers access full text "paywalled articles." We would like to draw particular attention to the usefulness of ORR. Although ORR does not immediately provide access and is a laborious procedure which depends on many factors that vary between individuals, ORR are quite efficient for obtaining full text "paywalled articles" (*Kanthraj, 2010*). Applied to our set of "paywalled articles" they allow free access to the full texts of 31 (26.96%) of 115, which is not negligible. An effort should be made to make them more widely known, particularly in less privileged countries, where they are the least used (*Burrows, 1996*). Another way to promote the use of ORR would be to have publishers and Pubmed make an extra effort to communicate, free of charge, corresponding authors' email addresses (only 37 emails were found on the PubMed website, and 81 were accessible free of charge on publisher websites). Indeed, some publishers disclose corresponding authors' email addresses only on the full text of articles attainable only through subscription, making it impossible to use ORR. Widely used globally (*Bohannon, 2016*), Sci-Hub raises many questions about the future of scientific publication (*Russell & Sanchez, 2016*; *McNutt, 2016*; *Machin-Mastromatteo, Uribe-Tirado & Romero-Ortiz, 2016*; *Strielkowski, 2017*). Our results showed that using Sci-Hub makes it possible to drastically reduce the inequalities of access to "paywalled articles" in the institutions studied, in terms of number of accessible full texts and the money spent to buy them on publishers' websites. As already pointed out by *Bendezú-Quispe et al. (2016)* and *Machin-Mastromatteo, Uribe-Tirado & Romero-Ortiz (2016)*, our data show that Sci-Hub can help clinicians in ophthalmology working in less privileged institutions or countries worldwide by allowing them to obtain essential information and respond appropriately to patient care needs. Without this option, they would not be able to cope with the demands of their profession. Nonetheless, as shown by *Bohannon (2016)* and *Machin-Mastromatteo, Uribe-Tirado & Romero-Ortiz (2016)*,

Sci-Hub is not only used in less privileged countries, and a correlation has been shown between the number of downloads per 1,000 inhabitants on Sci-Hub and the GDP per inhabitant (*Greshake, 2016*). As examples, the United States is the fifth largest downloader after Russia, and a quarter of the Sci-Hub requests came from the 34 members of the Organization for Economic Co-operation and Development, the wealthiest nations with, as shown in this study, the best journal access. The use of Sci-Hub by these countries can be defined as a use "by convenience" rather than necessity (*Bohannon, 2016*; *Hoy, 2017*; *Lawson, 2017*), and could easily decrease by improving library interfaces (*Faust, 2016*). In the long term, Sci-Hub might disrupt the whole system of academic publishing because it harms publishers due to the lost profits generated by its use (*Strielkowski, 2017*). In order to limit such losses, publishers could be tempted to increase their subscription rates (*Russell & Sanchez, 2016*), which have been steadily increasing in recent years (*Hoy, 2017*; *Himmelstein et al., 2018*), which may lead to cancellations or reduction of subscriptions by institutions worldwide (*Schiermeier & Mega, 2017*). This would have the effect of further accentuating inequalities. Publishers could also be tempted to further generalize "gold OA" (authors pay article processing charges (APC) to publish their articles) as the default publication model (*Novo & Onishi, 2017*; *Strielkowski, 2017*). This model is not beyond criticism because it favors the existence of predatory publishers (*Beall, 2012*). It also generates inequalities between authors who have funds to pay APC (whose cost is often prohibitive) and those who do not (*Danda, 2014*), despite many publishers providing discounts at the request of the authors or according to their geographical location. Some publishers have reacted by bringing lawsuits against Sci-Hub to shut down the site (*Kemsley, 2017*; *Greco, 2017*). So far, these attempts have been unsuccessful, and it is likely that future attempts will lead to the same outcome (*Hoy, 2017*). It now seems that Sci Hub will cease to operate only if and when the conditions that make it essential disappear (*Lawson, 2017*), i.e., that access to the articles through legal channels will not be as unequal as we have shown in this study. To achieve this, in addition to the suggestion mentioned above, several paths can be taken: including more countries in the Hinari program (particularly upper-middle income countries) (*Bendezú-Quispe et al., 2016*); promoting Green OA (self-archiving in OA repositories) by setting up institutional or national OA policies (*Kirsop & Chan, 2005*; *Machin-Mastromatteo, Uribe-Tirado & Romero-Ortiz, 2016*), (e.g., as it has been done recently in France) (*Boudry & Durand-Barthez, 2017*); and implementing subscription–based access for a reasonable price. This has already been done in other areas, particularly in the field of music, "offering individual subscription-based access to all articles from all imaginable databases for a price that most scientists in any corner of the world could afford" (*Strielkowski, 2017*).

## CONCLUSIONS

Regardless of the solutions chosen, it is urgent that the scientific community as well as decision-makers, mobilize effectively to limit these inequalities of access to scientific "paywalled articles" in order to solve this problem which has persisted for far too long, and finally free researchers from this daily dilemma: Sci-Hub or not Sci-Hub?

**Warning:** "Sci-Hub does not restrict itself to only openly licensed content. Instead, it retrieves and distributes scholarly literature without regard to copyright regimes. Readers should note that, in many jurisdictions, use of Sci-Hub may constitute copyright infringement. Users of Sci-Hub do so at their own risk. This study is not an endorsement of using Sci-Hub, and its authors and publishers accept no responsibility on behalf of readers. There is a possibility that Sci-Hub users—especially those not using privacy-enhancing services such as Tor—could have their usage history unmasked and face consequences, both legal or reputational in nature." (*Himmelstein et al., 2018*).

## ACKNOWLEDGEMENTS

The authors would like to thank Pr. Martine Jager (Leiden University Medical Center) for her help in finding a collaborator in the Netherlands and Chloé Rousseau and Dr. Florian Naudet for their help for reviewing the statistical analysis. Except for Boudry C. and Mouriaux F., all authors are listed alphabetically.

### Funding

The authors received no funding for this work.

### Competing Interests

The authors declare that they have no competing interests.

### Author Contributions

- Christophe Boudry conceived and designed the experiments, performed the experiments, analyzed the data, prepared figures and/or tables, authored or reviewed drafts of the paper, approved the final draft.
- Patricio Alvarez-Muñoz performed the experiments, authored or reviewed drafts of the paper, approved the final draft.
- Ricardo Arencibia-Jorge performed the experiments, authored or reviewed drafts of the paper, approved the final draft.
- Didier Ayena performed the experiments, authored or reviewed drafts of the paper, approved the final draft.
- Niels J. Brouwer performed the experiments, authored or reviewed drafts of the paper, approved the final draft.
- Zia Chaudhuri performed the experiments, approved the final draft.
- Brenda Chawner performed the experiments, approved the final draft.
- Emilienne Epee performed the experiments, approved the final draft.
- Khalil Erraïs performed the experiments, approved the final draft.
- Akbar Fotouhi performed the experiments, authored or reviewed drafts of the paper, approved the final draft.
- Almutez M. Gharaibeh performed the experiments, approved the final draft.
- Dina H. Hassanein performed the experiments, approved the final draft.

- Martina C. Herwig-Carl performed the experiments, authored or reviewed drafts of the paper, approved the final draft.
- Katherine Howard performed the experiments, authored or reviewed drafts of the paper, approved the final draft.
- Dieudonne Kaimbo Wa Kaimbo performed the experiments, authored or reviewed drafts of the paper, approved the final draft.
- Patricia-Ann Laughrea performed the experiments, approved the final draft.
- Fernando A. Lopez performed the experiments, approved the final draft.
- Juan D. Machin-Mastromatteo performed the experiments, authored or reviewed drafts of the paper, approved the final draft.
- Fernando K. Malerbi performed the experiments, authored or reviewed drafts of the paper, approved the final draft.
- Papa Amadou Ndiaye performed the experiments, approved the final draft.
- Nina A. Noor performed the experiments, approved the final draft.
- Josmel Pacheco-Mendoza performed the experiments, approved the final draft.
- Vasilios P. Papastefanou performed the experiments, approved the final draft.
- Mufarriq Shah performed the experiments, authored or reviewed drafts of the paper, approved the final draft.
- Carol L. Shields performed the experiments, authored or reviewed drafts of the paper, approved the final draft.
- Ya Xing Wang performed the experiments, approved the final draft.
- Vasily Yartsev performed the experiments, approved the final draft.
- Frederic Mouriaux conceived and designed the experiments, analyzed the data, prepared figures and/or tables, authored or reviewed drafts of the paper, approved the final draft.

## Data Availability

All data are available in the Supplemental Files.

## Supplemental Information

Supplemental information for this article can be found online at http://dx.doi.org/10.7717/peerj.7850#supplemental-information.

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
