# Peer review of "Worldwide inequality in access to full text scientific articles: the example of ophthalmology"

_PeerJ, doi:10.7717/peerj.7850_

## Round 0.1 · original submission · Minor Revisions

As you can see, both reviewers have only minor suggestions for the presentation of the data and methods. I would not anticipate any major problems in implementing these recommendations.

·

Basic reporting

See general comments. Article is well written in professional English. Some supplemental data files use French column names and documentation, although I imagine most users can deal with that.

Experimental design

See general comments.

Validity of the findings

See general comments.

Additional comments

The study compares fulltext access to 115 toll-access ophthalmology articles from 27 institutions around the world. In order to recruit participating institutions at a global scale, the organizers used an intriguing crowdsourcing approach by emailing corresponding authors of recent ophthalmology publications. The study found that alternative access methods (green OA combined with "online reprint request", i.e. requesting the PDF from an author) were able to access an average of 64% of the toll-access articles. Meanwhile, Sci-Hub provided access to 94% of the toll-access articles.

This study is impactful as a comparative analysis of institutional fulltext access across the globe. However, the analyses and visualziations do not present the findings of the survey to the fullest possible extent. Specifically, I was surprised not to see any plots of access rates by continent, income category, and Hinari access. A simple dot plot with access rate on the x-axes and category on the y-axis would provide important comparative summaries of the findings. Scatterplots may also be an option for comparing multiple continuous varaibles, with extra dimensions encoded by point color, size, or other attributes.

In addition, I was interested in discussion of the access rate of Hinari. It seems that this study should provide an answer to the question: "What percent of toll-access ophthalmology articles does Hinari provide access to?"

The discussion seems to imply that these analyses are not performed because of the low number of countries as well as the unequal selection. However, I still think it is important to visaualize access rates versus different factors, even in the absense of striking differences.

The study does not provide confidence intervals for proportions. There are many [existing, widely-implemented options](https://www.statsmodels.org/0.9.0/generated/statsmodels.stats.proportion.proportion_confint.html) to compute a confidence interval for a binomial proportion. In the [Sci-Hub Coverage Study](https://greenelab.github.io/scihub-manuscript/), we used the Jeffreys interval. It is essential that confidence intervals be reported, such that the reader can determine the uncertainty introduced by the small sample size of toll-access articles (n = 115). An example of a finding that demands a confidence interval is: "Used alone, Sci-hub allows the recovery of 108 full texts (93.9%) of 115." Presenting confidence intervals will likely illustrate that it would have been preferable to assess a larger number of toll-access articles (perhaps on the order of 500 articles). I do think the small sample size is a shortcoming of this study, but understand that increasing the sample size may not be possible at this stage. Confidence intervals are important to convey the uncertainty due to low sample size. It would be ideal to also present confidence intervals for the mean of access rates accross institutions. However, this is more complicated and may require a meta-analysis based approach.

Personally, I think "online reprint request" should not be grouped with Green OA availability into a single alternative access category. Specifically, ORR does not immediately provide access and is a laborious proceedure. It is also highly dependent on many factors that vary between invididuals. It is difficult to assess and control. So on one hand, it's helpful that the study analyzed ORR. On the other hand, it is not a reliable or controled method of article access. I understand that it may be too late to change this aspect of the study, but wanted to comment on it.

I wanted to point the authors attention to the 2016 study titled "[Correlating the Sci-Hub data with World Bank Indicators and Identifying Academic Use](https://doi.org/10.15200/winn.146485.57797)." I am not sure whether the authors were aware of this study, but it correlates Sci-Hub downloads to socioeconomic factors. These findings may provide interesting context for the fulltext access rates observed in this study.

In conclusion, the present work investigates a timely and important topic. The study design is innovative and reliable. The available of supplemental data is comprehensive. The main areas that need improvement are visualization of the findings and presentation of uncertainty.

·

Basic reporting

The manuscript is well structured and well written and I have only minor comments:

- The main body frequently refers to Sci-Hub being illegal. My understanding is that the legality of Sci-Hub depends on the jurisdiction in which the user accesses its services. The warning that's cited from Himmelstein et al 2018 in lines 381ff is more nuanced in that it points this out.
Unless the authors checked that using Sci-Hub is indeed illegal in all countries surveyed in this study, I'd encourage them to adopt a similarly nuanced language in the rest of the manuscript. If the authors checked and Sci-Hub is indeed illegal in all countries included in this study, it would be great if this could be pointed out too.

- I hate being the person who suggests adding one of their papers as a citation, but maybe http://dx.doi.org/10.15200/winn.146485.57797 could be useful to cite. In it I investigate global usage of Sci-Hub on a per-country level and correlate it with indicators such as GDP. This seems to fit well with the theme of this manuscript. But of course that addition is just a suggestion and by no means mandatory! :-)

- In some cases there are spaces missing to separate citations from the text (e.g. lines 105-108). It would be good to double check the references to see that they their formatting in the main body of the manuscript is correct.

Experimental design

Overall, the experimental design is well done, with a clearly outlined research question and appropriate methods to investigate it. The authors also do a great job in pointing out the limitations of their comparatively small random sample (200 journal articles out of 11,103 articles matching their search criteria). Given the large amount of manual labor involved in trying to access the paywalled proportion of these, the size of the subset is defensible.

The decision to limit the search to articles published in 2017-2018 could be a bit better motivated though. As the authors state (line 230f), recent articles are embargoed and will be found less frequently in the alternative access routes for that reason. My feeling is that sampling a bit older articles would avoid this issue and lead to numbers which are more 'stable' in time. I assume the focus on recent articles is done deliberately, as it's assumed that those articles are of most value to researchers who look for the latest findings to inform their own work. Pointing this out more clearly would be helpful.

Validity of the findings

All data are provided and the conclusions are well stated, supported and linked to the data and research questions. One minor thing:

The abstract states 'We found that complementary use of “alternative ways” and Sci-Hub leads to 95.5% of full text “paywalled articles”, and also divides by 14 the extra costs needed to obtain all full texts on publishers’ websites using pay-per-view.'

In the main body of the manuscript all I could find is 'Interestingly, complementary use of Sci-Hub reduces the average extra cost of unavailable full texts even more, dividing the maximum extra cost of unavailable full texts by 12.9'.

I could not fully understand whether this is based because of using different baselines for the calculations or because of a typo. It would be good to clear this up, as I couldn't find any reference to the 'dividing the cost by 14' anywhere but the abstract.

Additional comments

This is a really interesting manuscript about the impact of paywalls and alternative ways of accessing paywalled content. I think it will be a valuable addition for many people working in the field. All my comments above were really just minor changes that could make the argument a bit stronger. Well done!

---

## Round 0.2 · Minor Revisions

As you can see, only two minor issues need to be addressed before we can publish your work. I think both the suggestion wrt Fig. axis and font as well as the one wrt confidence intervals and margins of error make sense and should be relatively easy to address.

·

Basic reporting

See general comments

Experimental design

See general comments

Validity of the findings

See general comments

Additional comments

The authors addressed most of comments from my review of version 1.

The addition of Figure 2 is a major improvement. Figure 2b provides a striking finding: outside of the high-income bracket for countries, institutional + Hinari access is often below 50%. The two low income countries have particularly bad access. These findings suggests programs like Hinari are not succeeding in providing widespread access to countries that are not in the top income bracket.

I do have some remaining issues, regarding Figure 2 and the calculation of confidence intervals.

In Figure 2, rather than have the y-axis represent number of accessed articles out of 115, it would be clearer to plot the percent of articles accessed, such that the y-axis goes from 0% to 100%. The font size of the y-axis labels should be increased (currently they are practically illegible).

Regarding the confidence intervals / margins of error, I believe there are problems. The rebuttal letter states:

> In our case (sample size of 200), the confidence level is 85% (with a margin of error of 5%). We added at the end of Table 2, the confidence levels corresponding to the mean values. We also added, the sentence in Materials and Methods: “Confidence intervals were calculated taking into consideration the sample size studied (200), with a margin of error of 5%.”

I haven't verified the first quoted sentence, but let's assume that it is correct. This would mean that the _maximum margin of error_ is ±5% for a proportion with a sample size of 200 at 85% confidence. This statement is close to being relevant, except that most proportions reported in the manuscript have a sample size of 115 and not 200. I'd recommend computing the maximum margin of error with 95% confidence for a proportion with sample size of 115. This value could then be reported in the manuscript (not just the rebuttal letter) because it provides an upper bound on the margin of error for any reported proportion with sample size 115 (at the standard 95% confidence). In short, there are three variables here: the sample size (115), the confidence level (generally 95%), and then the margin of error / confidence interval. Usually, you choose a confidence level (generally 95%) and then calculate the margin of error **or** the confidence interval.

Note that the maximum margin of error is the radius of the largest-possible confidence interval given the sample size. The Wikipedia description is [not too dense](https://en.wikipedia.org/w/index.php?title=Margin_of_error&oldid=904288519#Maximum_and_specific_margins_of_error). In other words, for any specific proportion you can report a confidence interval. For any possible proportion of a given denominator (sample size), you can report a margin of error. For example, a (maximum) margin of error applies to any proportion computed from 115 samples. On the other hand, a confidence interval applies to a specific proportion like 70 / 115 and will generally be tighter than the maximum margin of error at the same confidence level.

Now let's turn our attention to "Confidence intervals were calculated taking into consideration the sample size studied (200), with a margin of error of 5%." First, the sample size of the proportions is 115, not 200. Second, the confidence interval and margin of error are the same thing: they are both ways of summarizing the output of an uncertainty assessment. Perhaps the authors mean "at 95% confidence" rather than "margin of error of 5%"?

In Table 2, the authors add a confidence interval row with the following caption: "confidence intervals were calculated considering the sample size studied (200), with a margin of error of 5%." Note that this description makes the same mistakes as the sentence added to Methods. Furthermore, the confidence intervals are presented (in the rebuttal letter only) as "corresponding to the mean values". However, the mean proportions across institutions cannot be computed with simple confidence intervals for proportions because they combine many individual proportions. The authors seem aware of this when they write: "Concerning the confidence intervals for the mean access rates across institutions, as mentioned, it is a more complicated approach which was not planned in our initial protocol and seems difficult to initiate now." Therefore, it doesn't make sense to report a margin of error / confidence interval for the mean proportion.

Without more advanced statistical expertise, I see a couple of options for the authors. The easiest option would be to report the maximum margin of error for a proportion with 115 samples at 95% confidence. Another option would be to report confidence intervals for each individual proportion (like 77 / 115), but not for the mean of proportions because that is more complicated. Both options can coexist. Hopefully, my explanation will help the authors determine how to proceed.

·

Basic reporting

no comment

Experimental design

no comment

Validity of the findings

no comment

Additional comments

The authors have fully addressed all my comments and concerns regarding the experimental design and validity and also clarified the language in the suggested places. I recommend accepting this manuscript in its revised version.

---

## Round 0.3 · accepted · Accept

Congratulations, the final reviewer accepted your manuscript.

·

Basic reporting

no comment

Experimental design

no comment

Validity of the findings

no comment

Additional comments

The revised manuscript addresses my remaining concerns. The margin of error is now only referenced in a single location as "9,14%". I presume this value should use a decimal separator of a period not a comma for consistency in the published version, i.e. "9.1%". I can confirm that ±9.1% is the proper maximum margin of error for a proportion with a sample size of 115.